# *TNFSF9* Is Associated with Favorable Tumor Immune Microenvironment in Patients with Renal Cell Carcinoma Who Are Treated with the Combination Therapy of Nivolumab and Ipilimumab

**DOI:** 10.3390/ijms25137444

**Published:** 2024-07-06

**Authors:** Bunpei Isoda, Shuya Kandori, Tomokazu Sazuka, Takahiro Kojima, Satoshi Nitta, Masanobu Shiga, Yoshiyuki Nagumo, Ayumi Fujimoto, Takayuki Arai, Hiroaki Sato, Bryan J. Mathis, Chia-Ling Wu, Yi-Hua Jan, Tomohiko Ichikawa, Hiroyuki Nishiyama

**Affiliations:** 1Department of Urology, Institute of Medicine, University of Tsukuba, Tsukuba 305-8577, Ibaraki, Japan; bunpeiisoda@gmail.com (B.I.); n.satoshi19880320@gmail.com (S.N.); baseball4420002001@yahoo.co.jp (M.S.); nagumoyoshiyuki@gmail.com (Y.N.); nishiuro@md.tsukuba.ac.jp (H.N.); 2Department of Urology, Graduate School of Medicine, Chiba University, Chiba 263-8522, Chiba, Japan; tomo1ata2@yahoo.co.jp (T.S.); ayuroom@gmail.com (A.F.); takataka50423@gmail.com (T.A.); hhiirrooaakkii14@yahoo.co.jp (H.S.); tomohiko_ichikawa@faculty.chiba-u.jp (T.I.); 3Department of Urology, Aichi Cancer Center, Nagoya 464-8681, Aichi, Japan; t.kojima@aichi-cc.jp; 4International Medical Center, University of Tsukuba Affiliated Hospital, Tsukuba 305-8576, Ibaraki, Japan; bmathis@md.tsukuba.ac.jp; 5ACT Genomics, Co., Ltd., Taipei 114, Taiwan; jalinwu@actgenomics.com (C.-L.W.); isaacjan@actgenomics.com (Y.-H.J.)

**Keywords:** metastatic renal cell carcinoma, *TNFSF9*, biomarker, immune checkpoint inhibitor combination therapy

## Abstract

Combination therapy of nivolumab and ipilimumab (NIVO + IPI) for metastatic renal cell carcinoma (mRCC) has shown efficacy, but approximately 20% of patients experience disease progression in the early stages of treatment. No useful biomarkers have been reported to date. Therefore, it is desirable to identify biomarkers to predict treatment responses in advance. We examined the tumor microenvironment (TME)-related gene expression in mRCC patients treated with NIVO + IPI, between the response and non-response groups, using tumor tissues, before administering NIVO + IPI. In TME-related genes, *TNFSF9* expression was identified as a candidate for the predictive biomarker. Its expression discriminated between the response and non-response groups with 88.89% sensitivity and 87.50% specificity (AUC = 0.9444). We further analyzed the roles of *TNFSF9* in TME using bioinformatics from The Cancer Genome Atlas (TCGA) cohort. An adaptive immune response was activated in the *TNFSF9*-high-expression tumors. Indeed, T follicular helper cells, plasma B cells, and tumor-infiltrating CD8^+^ T cells were increased in the tumors, which indicates the promotion of humoral immunity due to enhanced T-B interactions. However, as the number of regulatory T cells (Treg) increased in the tumors, the percentage of dysfunctional T cells also increased. This suggests that not only PD-1 but also CTLA-4 inhibition may have suppressed Treg activation and improved the therapeutic effect in the *TNFSF9* high-expression tumors. Therefore, *TNFSF9* may predict the therapeutic efficacy of NIVO + IPI for mRCC and allow more appropriate patient selection.

## 1. Introduction

The CheckMate-214 trial has demonstrated the efficacy of combination therapy with nivolumab and ipilimumab (NIVO + IPI) in patients with metastatic renal cell carcinoma (mRCC) [1]. Especially, NIVO + IPI bring durable responses for some patients. However, approximately 20% of patients experience disease progression in the early stages of NIVO + IPI. Therefore, there is a need to identify biomarkers to predict treatment responses in advance.

The biomarker analysis of the CheckMate-214 trial investigated genomic and transcriptomic biomarkers and showed the association between these biomarkers and survival outcomes in patients treated with NIVO + IPI [2]. The PD-L1 expression level was significantly associated with better progression-free survival (PFS) or overall survival (OS) in mRCC patients who were treated with NIVO + IPI. On the other hand, genomic features such as tumor mutation burden (TMB) and tumor indel burden (TIB) were not associated with responses to NIVO + IPI. Neither PFS nor OS was significantly associated with the mutation status of the hallmark genes except *PBRM1* [2]. Moreover, gene expression signatures did not independently predict PFS or OS in these patients. This study suggests that combining multiple gene expression signatures may predict response to NIVO + IPI. However, no effective biomarkers have been identified to predict the therapeutic efficacy of NIVO + IPI for mRCC in clinical practice.

We analyzed the tumor microenvironment (TME)-related genes to predict the responses to NIVO + IPI for mRCC patients using pretreated primary tumor tissues. *TNFSF9* expression was significantly upregulated in the responder group compared to the non-responder groups. The transcriptomic analysis using The Cancer Genome Atlas (TCGA)–kidney renal clear cell carcinoma (KIRC) cohort revealed an increase in dysfunctional T cells with regulatory T cells (Treg) in the high-*TNFSF9*-expression tumors, despite the activation of an adaptive immune response. This suggests that not only PD-1 but also CTLA-4 inhibition may have suppressed Treg activation and improved the therapeutic effect for the high-*TNFSF9*-expression tumors.

## 2. Results

### 2.1. Patient Characteristics

Of 17 total patients, 8 (8 male, no female) were classified into the responder group, and 9 (8 male, 1 female) were classified into the non-responder group. Patient characteristics are summarized in Table 1. The median ages of responders and non-responders were 70 years (60–73 years) and 69 years (52–81 years). Regarding IMDC risk, among the responders, seven patients were in the intermediate risk group and one patient was in the poor risk group; meanwhile, among the non-responders, seven patients were in the intermediate risk group and two patients were in the poor risk group. No patient had brain metastases. There were no significant differences in patient characteristics between each group.

### 2.2. PD-L1, MSI, and TMB Status

There were PD-L1-positive tumors (37.5%) in responders and 1 PD-L1-positive tumor (11.1%) in non-responders, but there were no significant differences between groups (*p* = 0.2941). None of them showed high microsatellite instability or tumor mutation burden in our cohort (Table 2).

### 2.3. Gene Mutation Status

In this patient cohort, oncoprint analyses were created by focusing on variants that were presumed to be oncogenic drivers in OncoKB among the genes that were found to be mutated in this study (Figure 1). The most frequently mutated genes were *VHL* (47%), *PBRM1* (41%), and *SETD2* (24%). In terms of the percentage of these gene mutations, they are slightly more common in *SETD2*, but the rest are comparable to those reported in the past [3,4,5,6]. There were no significant differences in the expression frequencies of these genes between responders and non-responders (Table 3).

### 2.4. Tumor Microenvironment Profiling for the Immunotherapy Biomarker Exploration

Almost all differentially expressed genes (DEGs) were upregulated in the response group compared to the non-response group (Figure 2A). DEGs were defined as genes with a fold change of 2-fold or more and a *p*-value of 0.05 or less. *TNFSF9*, *IDO1*, *TIGIT*, *ICOS*, *CCL5*, *ITK*, *FASLG*, *CRTAM*, *HLA-A*, *GNLY*, and *INFG* were extracted as DEGs (Figure 2B). The most significant difference and fold change were seen in *TNFSF9* (tumor necrosis factor superfamily member 9), also known as *4-1BBL* or *CD137L*. *TNFSF9* is expressed primarily in antigen-presenting cells and has been shown to function as a potent co-stimulatory molecule [7]. Furthermore, the heatmap for these DEGs shows that the expression pattern changes with treatment response (Figure 2C).

Also, when ROC analysis is performed for each gene, there are 12 genes with *p*-values of less than 0.05. Of these, three have an area under the curve (AUC) greater than 0.8: *TNFSF9*, *TAP1*, and *CD8A* (Figure 3A). Interestingly, significant differences in each gene show the smallest *p*-value for *TSFNF9* (*p* = 0.0010) (Figure 3B). The AUC is also the largest for *TNFSF9* (AUC = 0.9444), whereas that of PD-L1 is lower (AUC = 0.75) (Figure 3A,C). These results suggest that TSFNF9 is a potential predictive biomarker for immunotherapy in patients with mRCC.

### 2.5. Characteristics of Tumor Immune Microenvironment in High-TNFSF9-Expression Tumors

To characterize the TIME of high-*TNFSF9* tumors in the KIRC cohort, we calculated the stromal score, immune score, and ESTIMATE score. The ESTIMATE score is a method that estimates the percentage of stromal cells and immune cells in a tumor sample using gene expression signatures. High expression of *TNFSF9* was associated with significantly higher immune scores and higher ESTIMATE scores, but not stromal scores (Figure 4A). Therefore, tumors with high *TNFSF9* expression were characterized by higher tumor purity and higher immune cell infiltration.

The analysis of DEGs using Metascape revealed that gene pathways related to adaptive immune responses, response to bacteria, and NABA MATRISOME (a collection of extracellular matrix-associated proteins and related factors) [8] were upregulated in *TNFSF9*-high-expression tumors (Figure 4B). In addition, we analyzed the population of immune cells infiltrating tumor cells using TIMER, finding that B cells, CD8^+^ T cells, and myeloid dendritic cells were significantly increased in the high-*TNFSF9*-expression group (Figure 4C). These findings suggest that tumors with high *TNFSF9* expression have more antigen-presenting cells and occur at a certain level of acquired immune responses. It is also possible that in the *TNFSF9*-high-expression group, the immune response is better regulated by the inter-relationship between the extracellular matrix and immune cells [9].

### 2.6. Changes in T Cell Signatures via TNFSF9 Expression

An analysis of T cells via TIDE also showed significantly more CD8^+^ T cells and significantly higher levels of interferon-gamma in the *TNFSF9*-high tumors (Figure 5A,B). In addition, there were significantly fewer exclusionary T cells and more dysfunctional T cells in the *TNFSF9*-high tumors (Figure 5C,D), suggesting that T cell function may be more impaired in the tumors. However, the association between *TNFSF9* and *CD274* expression was not significant (Figure 5E).

The TIDE score is calculated using two mechanisms of immune evasion: the induction of T cell dysfunction in tumors with high infiltration of cytotoxic T cells and the inhibition of T cell infiltration in tumors with few cytotoxic T cells [10]. In general, patients with high TIDE scores are less likely to respond to immune checkpoint inhibitor (ICI) monotherapy (CTLA4 inhibitors or PD-1 inhibitors), and the predominance of high TIDE scores in the *TNFSF9*-high tumor suggests that ICI monotherapy may be less effective (Figure 5F,G).

### 2.7. Characteristics of Tumor-Infiltrating Immune Cells in High-TNFSF9-Expression Tumors

Populations of tumor-infiltrating immune cells by *TNFSF9* expression were examined using the CIBERSORT method (Figure 6). Plasma B cell, CD8^+^ T cell, T follicular helper (T_FH_) cell, regulatory T (Treg) cell, and M1 Macrophage levels were significantly increased in the *TNFSF9*-high-expression tumor. In contrast, activated the mast cell level is significantly decreased. The increase in Treg cells may cause dysfunction of T cells in *TNFSF9*-high-expression tumors (Figure 5C and Figure 6). On the other hand, the increase in T_FH_ cells and plasma B cells suggests that T-B interaction leads to humoral immunity for ccRCC in *TNFSF9*-high-expression tumors (Figure 6). These findings suggest that both adaptive immune responses and the dysfunction of T effector cells are promoted in high-*TNFSF9* tumors.

## 3. Discussion

The prognosis of mRCC has improved in the immunotherapy era. ICIs have emerged as the mainstay and backbone of the treatment strategy for most patients with mRCC. Combination therapy with ICIs—NIVO + IPI or ICI plus tyrosine kinase inhibitor (TKI) combination therapy (IO + TKI)—is the first-line treatment option for mRCC in the NCCN guidelines [11]. Comparing NIVO + IPI to IO-TKI, the duration of response (DOR) is about 2 years for IO-TKI [12,13], whereas NIVO + IPI has proven a longer DOR in patients with mRCC [14]. However, NIVO + IPI have been shown to cause about 20% disease progression in the early stages of treatment [1]. Therefore, it is desirable to find the biomarkers that predict therapeutic responses to NIVO + IPI and to identify suitable patients for treatment. Cytoreductive nephrectomy (CN) is also a keystone treatment strategy for mRCC. Indeed, several clinical trials are ongoing to elucidate the role of CN in the immunotherapy era [15]. As such, predictive biomarkers might bring significant insight into the appropriate patient selection for CN.

Biomarkers have been previously reported to predict the efficacy of systemic therapy for mRCC, as in the example of transmembrane protein mucin-1 (MUC1), which was highlighted as a candidate biomarker for antiangiogenic treatment [16]. However, reliable predictive biomarkers for NIVO + IPI remain unidentified for clinical practice. In our current study, we found that *TNFSF9* expression is most predominantly elevated in the response group of NIVO + IPI. Moreover, its expression discriminated between the response group and the non-response group with 88.89% sensitivity and 87.50% specificity (AUC = 0.9444) in our cohort. This suggests that *TNFSF9* is a potential biomarker to predict the efficacy of NIVO + IPI for patients with mRCC.

*TNFSF9* is expressed not only on antigen-presenting cells but also on non-immune cells [17]. Its receptor, tumor necrosis factor receptor superfamily member 9 (TNFRSF9), is a member of the TNF receptor superfamily and has been identified as a co-stimulatory molecule for T cells [18]. This receptor is expressed as a transmembrane protein on the cell surface and functions to receive and transmit signals to the expressing cells via ligand binding. It has been reported in monocytes to induce proliferation [19], differentiation [20], maturation, and the production of inflammatory cytokines [21], while inhibiting proliferation through apoptosis in T lymphocytes [22]. Therefore, these findings speculate that *TNFSF9*-*TNFRSF9* interaction affects TIME in mRCC patients treated with NIVO + IPI.

The relationship between *TNFSF9*-*TNFRSF9* and TIME has been reported in several studies. Cho et al. revealed that in non-small-cell lung cancer, reduced levels of TNFRSF9 expression in Tregs improve OS and responses to anti-PD-1 antibody immunotherapy. The authors speculated that TNFRSF9 is related to the enhanced immunosuppressive activity of Tregs in the tumor. However, they also reported that Treg expression levels did not correlate with response to anti-PD-1 antibody therapy in patients with malignant melanoma, using two independent cohorts [23]. On the other hand, Wu et al. revealed that *TNFSF9* expression in the tumor and CD8^+^ T cell infiltration were negatively correlated in pancreatic cancer [24]. Conversely, *TNFSF9* expression in RCC was positively associated with CD8^+^ T cell infiltration in this study. These findings may be attributed to the different roles of *TNFSF9*-*TNFRSF9* in the TIME according to cancer type.

The relationship between RCC and *TNFSF9* has not been previously reported. A gene pathway analysis using Metascape revealed that the adaptive immune response is promoted in *TNFSF9*-high tumors. Moreover, an analysis using CIBERSORT revealed that both T_FH_ cells and plasma B cells are increased in the tumors. These results suggest that T-B interactions lead to humoral immunity through the promotion of the adaptive immune system in *TNFSF9*-high tumors. However, our analysis also revealed that tumor-infiltrating CD8^+^ T cells, especially dysfunctional CD8^+^ T cells, are increased in the tumors. Many factors contribute to CD8^+^ T cell dysfunction, one of which has been shown to involve Tregs [25]. CTLA-4, expressed on Tregs, competes with CD28 for binding to CD86/80 on antigen-presenting cells, exerting inhibitory control over CD8^+^ T cell activation [26]. Anergy induction by Tregs leads to CD8^+^ T cell unresponsiveness, impairing their ability to mount an effective immune response. Furthermore, immunosuppressive factors, such as TGF-β and IL-10, released by Tregs, contribute to the immunosuppression of CD8^+^ T cells, inhibiting effector cytokine production and fostering an anti-inflammatory milieu [27]. Tregs also sequester IL-2, essential for T cell proliferation, thereby limiting its availability for CD8^+^ T cell expansion [28]. Indeed, the CIBERSORT analysis showed that Tregs were predominantly increased in *TNFSF9*-expressing tumors. Thus, the increase in Tregs may be the cause of the increase in dysfunctional CD8^+^ T cells. Moreover, the previous study demonstrated that *TNFSF9*-containing extracellular vesicles from cancer cells promote the immunosuppressive activity of Tregs in leukemia [29]. Therefore, these findings suggest that *TNFSF9* promotes T cell exhaustion via the activation of Tregs in the TIME of mRCC.

Nivolumab binds to PD-1 on T cells and blocks its interaction with tumor-expressed PD-L1, thereby inhibiting immune evasion and promoting T cell activation. However, tumor-infiltrating CD8^+^ T cells and Treg cells express PD-1 at similarly high levels, and the administration of anti-PD-1 antibodies may cause the inadvertent activation of Tregs and the enhancement of immunosuppression [30]. To address this, inhibiting receptors that are highly expressed on tumor-infiltrating Tregs can enhance therapeutic efficacy; CTLA-4 is one such receptor. Ipilimumab may play an important role in inhibiting CTLA-4, thereby reducing Treg activation and improving therapeutic efficacy [31,32]. Collectively, these findings support the rationale that combination therapy with NIVO + IPI might be effective for mRCC patients with *TNFSF9*-high tumors.

There are several limitations to the present study. First, we could not exclude the possibility of selection bias due to the retrospective design of the study. Moreover, the sample size is relatively small. As a consequence, only one female was included in the present study. Finally, patient backgrounds were heterogeneous, possibly influencing the results. Further studies would be necessary to rectify these limitations in the future.

## 4. Materials and Methods

### 4.1. Patients

We included 17 mRCC patients who were treated with NIVO + IPI at the University of Tsukuba or Chiba University hospitals from 2018 to 2020. Patients were divided into response and non-response groups. The responder group included patients with a complete or partial response as the best treatment response, and the non-responder group included patients with progressive disease.

### 4.2. DNA and RNA Isolation

A total of 6 biopsy and 11 surgical specimens were obtained from 17 mRCC patients. For analysis, 5 µm-thick slices were prepared from formalin-fixed paraffin-embedded (FFPE) blocks before total DNA and RNA were extracted from these samples with a RecoverAll™ Total Nucleic Acid Isolation Kit for FFPE (Thermo Fisher Scientific, Waltham, MA, USA) according to the manufacturer’s instructions.

### 4.3. ACTOnco Next-Generation Sequencing

The extracted genomic DNA was amplified using four pools of primer pairs that target the coding exons of the analyzed genes. Amplicons were ligated with barcoded adaptors. The quality and quantity of amplified libraries were determined using the fragment analyzer (AATI) and Qubit (Invitrogen, Thermo Fisher Scientific). Barcoded libraries were subsequently conjugated with sequencing beads using emulsion PCR and enriched using an Ion Chef system (Thermo Fisher Scientific) according to the Ion PI Hi-Q Chef Kit (Thermo Fisher Scientific) or Ion 540 Kit-Chef protocols (Thermo Fisher Scientific). Sequencing was performed on an Ion Proton or Ion S5 sequencer (Thermo Fisher Scientific).

Raw reads generated using the sequencer were mapped to the hg19 reference genome using the Ion Torrent Suite (version 5.10). Coverage depth was calculated using the Torrent Coverage Analysis plug-in. Single nucleotide variants (SNVs) and short insertions/deletions (INDELs) were identified using the Torrent Variant Caller plug-in (version 5.10). The coverage was down-sampled to 4000. OncoKB (version 20231206) was used to annotate every variant. Variants with coverage ≥25, allele frequency ≥ 5%, and actionable variants with allele frequency ≥2% were retained. This test provides uniform coverage of the targeted regions, enabling target base coverage at 100× ≥ 85% with a mean coverage ≥ 500×. Variants reported in the Genome Aggregation database r2.1.1 with >1% minor allele frequency (MAF) were considered polymorphisms. An ACT Genomics in-house database was used to determine technical errors. Clinically actionable and biologically significant variants were determined based on the published medical literature.

The copy number variations (CNVs) were predicted as follows: Amplicons with read counts in the lowest 5th percentile of all detectable amplicons and amplicons with a coefficient of variation ≥ 0.3 were removed. The remaining amplicons were normalized to correct the pool design bias. ONCOCNV was applied for the normalization of total amplicon number, amplicon GC content, amplicon length, and technology-related biases, followed by segmenting the sample with a gene-aware model. This method was also used for establishing the baseline of copy number variations from samples in the ACT Genomics in-house database [33].

### 4.4. ACT TME

ACT TME™ utilizes TaqMan™ OpenArray^®^ (Thermo Fisher Scientific) technology to measure the expression signature of 106 immune-related genes involved in antigen presentation, immune checkpoint regulation, immune cell population identification, and other modulators relevant to the tumor microenvironment. In terms of the standard workflow for using the TaqMan™ OpenArray^®^ instrument, FFPE RNA was first reverse-transcribed to cDNA with the GeneAmp PCR System 9700 (Thermo Fisher Science) before a pre-amplification step to ensure a sufficient template for qPCR reaction using a Veriti 96-Well Thermal Cycler (Thermo Fisher Science). Next, the mixture of sample and master mix was loaded onto a QuantStudio 12K Flex OpenArray chip (Thermo Fisher Science), which contained 48 subarrays, using a QuantStudio 12K Flex AccuFill System (Thermo Fisher Science). Finally, the sample-filled OpenArray chip was uploaded to a QuantStudio 12K Flex Real-Time PCR System (Thermo Fisher Science).

### 4.5. Use of Transcriptomic Data from the Cancer Genome Atlas

Gene expression datasets and clinicopathological data were downloaded from the TCGA-KIRC project (http://cancergenome.nih.gov) in 1 April 2020. The available transcriptomics data from 532 clear cell renal cell carcinoma (ccRCC) specimens whose sample type codes were registered as 01A were used.

### 4.6. Estimation of Stromal and Immune Cells in Tumors

The levels of infiltrating stromal and immune cells were estimated for each sample based on the gene expression profiles, utilizing the “Estimation of STromal and Immune cells in MAlignant Tumours using Expression data” (ESTIMATE) algorithm (https://bioinformatics.mdanderson.org/publicsoftware/estimate/ (accessed on 28 March 2022)) [34].

### 4.7. Metascape

Metascape (v3.5.20240101), an online bioinformatics tool, was employed for gene list enrichment analysis in this study. It facilitates the functional interpretation of gene sets by integrating multiple annotation sources (https://metascape.org/gp/index.html#/main/step1 (accessed on 28 March 2022)) [35].

### 4.8. Estimation of Tumor-Infiltrating Immune Cells in Tumors

The Tumor IMmune Estimation Resource 2.0 (TIMER2.0) (http://cistrome.org/TIMER/ (accessed on 28 March 2022)) [36] was used to estimate the population of tumor-infiltrating immune cells.

### 4.9. Evaluation of T Cell Signatures in Tumors

Tumor Immune Dysfunction and Exclusion (TIDE) scores were calculated in all samples (http://tide.dfci.harvard.edu/login/) [10]. The TIDE algorithm was used to estimate the projected response of each sample to anti-PD-1/PD-L1 and anti-CTLA4 immunotherapy based on gene expression profiles.

### 4.10. Statistical Analyses

All statistical analyses were performed using R4.3.1 (R Development Core Team, Vienna, Austria), JMP 10 software (SAS Institute, Cary, NC, USA), or GraphPad Prism8 (GraphPad Software, San Diego, CA, USA).

The significance of any differences between groups was assessed using Fisher’s exact test, Wilcoxon rank-sum test, or Mann–Whitney U test. Patients were divided into two groups, a low-expression or high-expression group, using the cutoff of median expression values. *p*-values < 0.05 were considered statistically significant.

## 5. Conclusions

Taken together, our results indicate that *TNFSF9* is a potential biomarker to predict the efficacy of NIVO + IPI for patients with mRCC. Moreover, a bioinformatics analysis suggests that *TNFSF9* is associated with a favorable TIME for combination therapy with NIVO + IPI in mRCC patients. Additional studies are needed to understand the detailed mechanisms of how *TNFSF9* affects the TIME in RCC, especially regarding Treg functions.

## Figures and Tables

**Figure 1 ijms-25-07444-f001:**
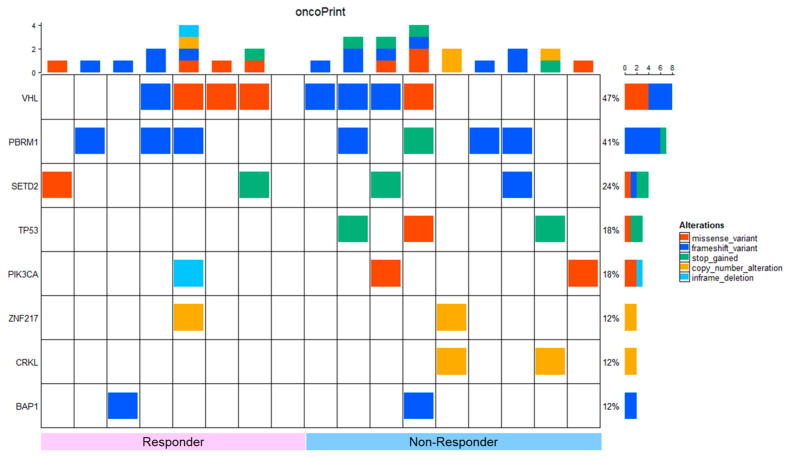
Oncoprint of gene variants in primary tumors between responders and non-responders in our cohort. Red indicates a missense variant, blue indicates a frameshift variant, green indicates a gain of stop codon, yellow indicates a copy number alteration, and light blue indicates an in-frame deletion.

**Figure 2 ijms-25-07444-f002:**
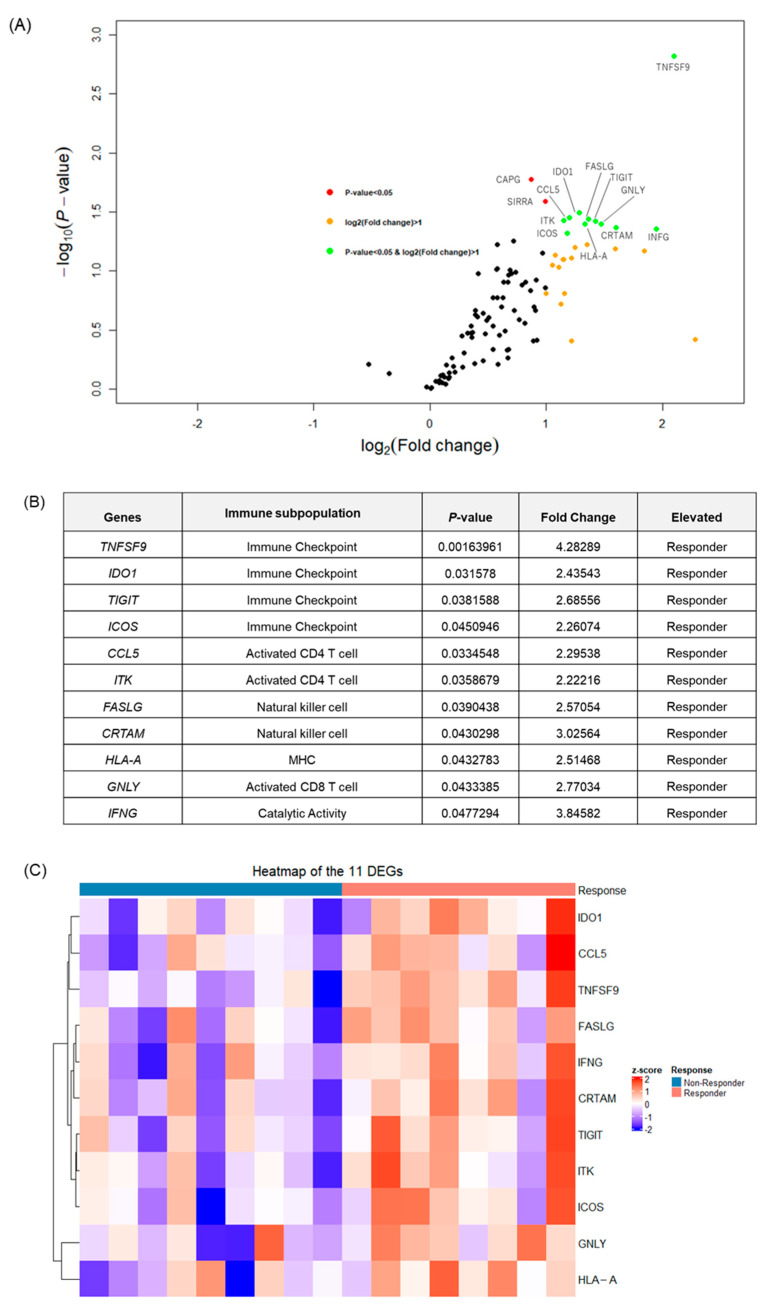
The differentially expressed genes (DEGs) in primary tumors between responders and non-responders. (**A**) A volcano plot showing the distribution of the DEGs between responders and non-responders. (**B**) A list of the significant DEGs between responders and non-responders. (**C**) A heatmap of the 11 significant DEGs. Shades of blue and red correlate with increasing or decreasing Z scores (from −2 to 2), with white representing a Z score of 0.

**Figure 3 ijms-25-07444-f003:**
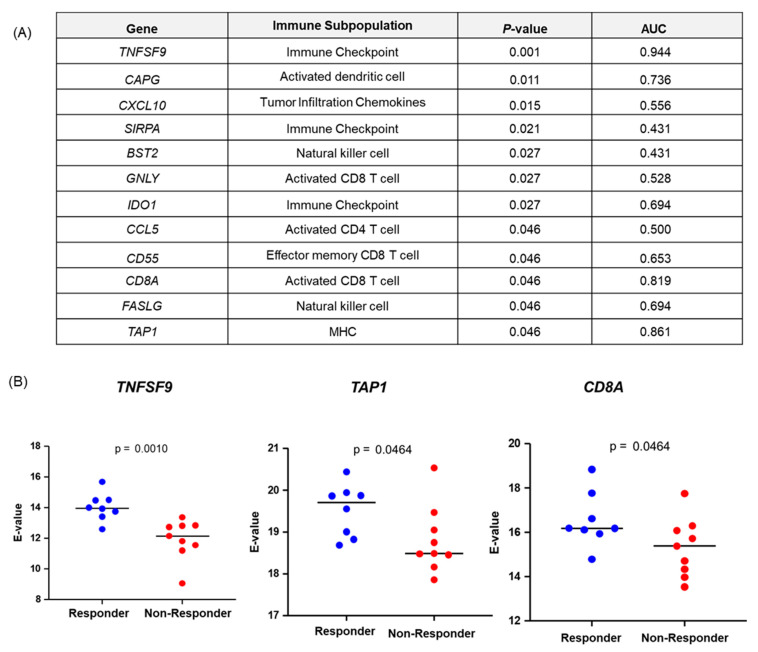
Receiver Operating Characteristics (ROC) analysis. (**A**) The list of potential biomarker genes. (**B**) The gene expression levels between responders and non-responders. (**C**) ROC curves of *TNFSF9 and PL-D1*, showing the sensitivity and specificity of the analysis regarding the *TNFSF9 and* PL-D1 predictions of NIVO + IPI response. AUC = area under curve.

**Figure 4 ijms-25-07444-f004:**
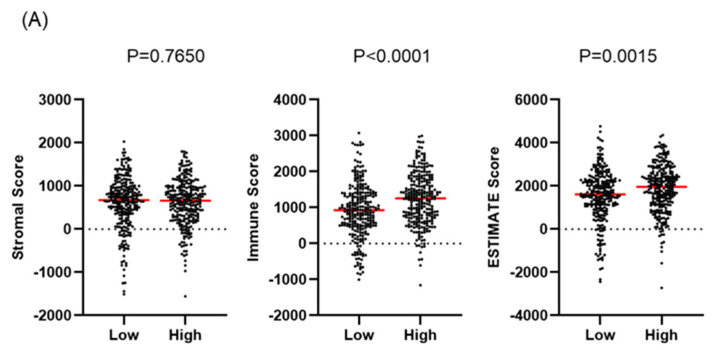
The evaluation of the tumor immune microenvironment between low- and high-*TNFSF9* tumors. (**A**) The stromal, immune, and ESTIMATE scores between high- and low-*TNFSF9* tumors were evaluated using the ESTIMATE algorithm. The red line is the median and the same thereafter. (**B**) The difference in Gene Ontology biological process categories between high- and low-*TNFSF9* tumors. (**C**) The proportion of tumor-infiltrating immune cells between high and low *TNFSF9* was evaluated using the TIEMR algorithm.

**Figure 5 ijms-25-07444-f005:**
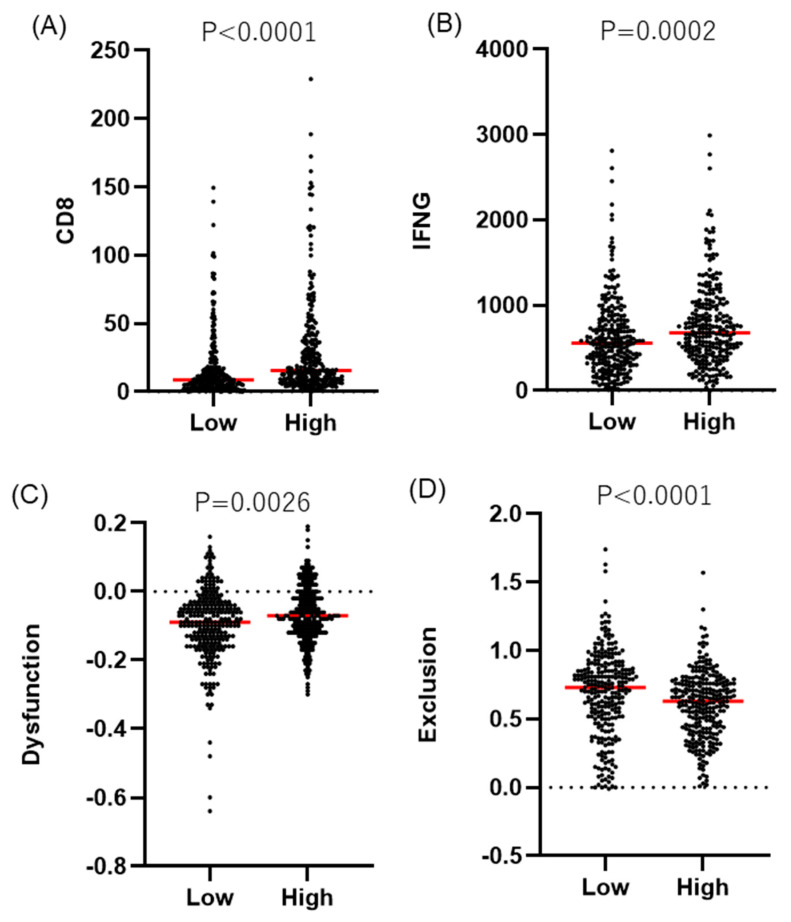
The evaluation of T cell signatures using TIDE between low- and high-*TNFSF9* tumors. The dashed line represents 0. (**A**) CD8^+^ T cell. (**B**) Interferon gamma. (**C**) Dysfunction of the T cell. (**D**) Exclusion of the T cell. (**E**) Expression status of CD274 (PD-L1). (**F**) TIDE score. (**G**) Response rate to immune checkpoint inhibitor monotherapy as predicted by the TIDE score.

**Figure 6 ijms-25-07444-f006:**
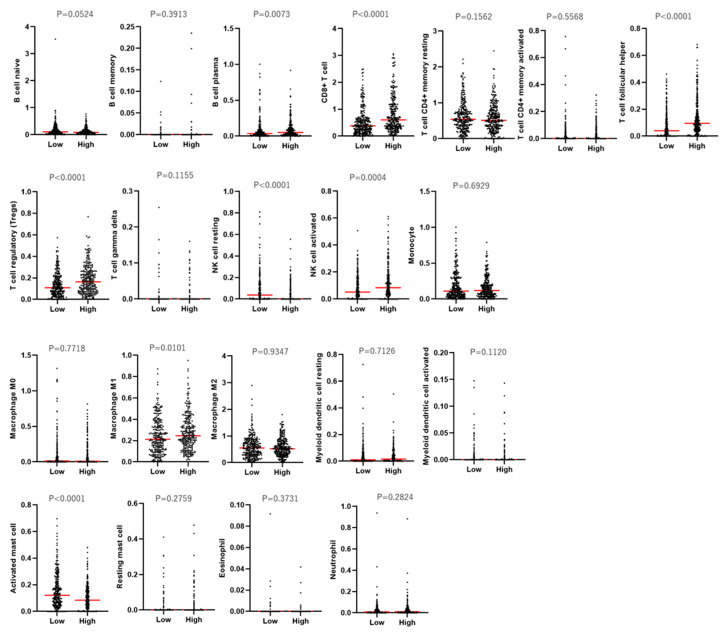
The evaluation of tumor-infiltrating immune cells using CIBERSORT-ABS between low- and high-*TNFSF9* tumors.

**Table 1 ijms-25-07444-t001:** Patient characteristics.

	Responder	Non-Responder
Median age (range)	70 (60–73)	69 (52–81)
Sex Male/Female	8/0	8/1
IMDC risk		
Intermediate	7	7
Poor	1	2
Previous nephrectomy	6	5
Pathology		
Clear cell renal cell carcinoma	7	7
Clear cell renal cell carcinoma with sarcomatoid	0	1
Papillary renal cell carcinoma	1	1
Sites of metastasis		
Lung	4	5
Bone	2	3
Liver	1	2
Brain	0	0
Lymph node	1	3

**Table 2 ijms-25-07444-t002:** PD-L1, MSI, and TMB status between responders and non-responders.

	Responder	Non-Responder	*p*-Value
PD-L1 (SP263 TC)—positive	3 (37.5%)	1 (11.1%)	0.2941
MSI-High	0 (0%)	0 (0%)	-
TMB			-
High	0 (0%)	0 (0%)	
Cannot be determined	3 (37.5%)	1 (11.1%)	

**Table 3 ijms-25-07444-t003:** Gene mutation status between responders and non-responders.

Gene	Responder(%)		Non-Responder(%)		*p*-Value
* **VHL** *	4	(50.0)	4	(44.4)	1.000
* **PBRM1** *	3	(37.5)	4	(44.4)	1.000
* **SETD2** *	2	(25.0)	2	(22.2)	1.000
* **BAP1** *	1	(12.5)	1	(11.1)	1.000
* **TP53** *	0	(0)	3	(33.3)	0.206

## Data Availability

The datasets generated during and/or analyzed during the current study are available from the corresponding author upon reasonable request.

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
