# Peer review of "TNFSF9 Is Associated with Favorable Tumor Immune Microenvironment in Patients with Renal Cell Carcinoma Who Are Treated with the Combination Therapy of Nivolumab and Ipilimumab"

_ijms, 2024, doi:10.3390/ijms25137444_

Round 1

Reviewer 1 Report

Comments and Suggestions for Authors

In this manuscript, Bunpei et al. tried to identify a biomarker to predict the treatment of mRCC with combination therapy of NIVO plus IPI. TNFSF9, also known as 4-1BB ligand, was identified as the potential candidate by comparing the response and no-response groups. The immune cells like plasma B cells and tumor-infiltrating CD8+ T cells were increased in the tumor microenvironment. Meanwhile, Treg, known as T cell inhibition function, was also increased. I believe this study is scientifically sound and interesting to readers. Several questions/comments as follows:

Among the 17 patients, there is only one female. Do you think gender will have an influence? Is there a bias during patient selection?

As shown in Figure 2A, why almost all genes were upregulated? Did you check the downregulated genes? 

Author Response

Reviewer 1
“Among the 17 patients, there is only one female. Do you think gender will have an influence? Is there a bias during patient selection?"

Thank you for your suggestion. The gender bias may be due to the small sample size in the present study. As you pointed out, it may affect the results. It is a limitation in the present study, so we have added some descriptions in the text.

Lines 283-287

There are several limitations in the present study. First, we could not exclude the possibility of selection bias due to the retrospective design of the study. Moreover, the sample size is relatively small. As a consequence, only one female was included in the present study. Finally, patient backgrounds were heterogeneous, possibly influencing the results. Further studies would be necessary to rectify these limitations in the future.

“As shown in Figure 2A, why almost all genes were upregulated? Did you check the downregulated genes?”

Thank you for pointing this out. We also analyzed the downregulated genes in the present study. As shown in Figure 2A, some genes were downregulated. However, the expression levels were not significantly changed.

Reviewer 2 Report

Comments and Suggestions for Authors

The study addresses an important gap in the therapeutic management of metastatic renal cell carcinoma (mRCC), focusing on the combination therapy of nivolumab and ipilimumab. Its main contribution is the identification of TNFSF9 gene expression as a potential biomarker for predicting treatment response. This is particularly novel as it highlights a specific gene that could significantly impact patient stratification and treatment planning, addressing a critical need given that about 20% of patients experience early disease progression under the current treatment regime.

Areas for Improvement:

  1. Sample Size: The study involves a relatively small cohort of 17 patients. Increasing the sample size could enhance the robustness of the findings and ensure they are more generalizable. If this si not possible please better describe how this limited number of patients could affect your findings. 
  2. Control Comparisons: While the article focuses on TNFSF9, it would be beneficial to compare its predictive power directly against other potential biomarkers for a more comprehensive evaluation. 
  3. Long-term Outcomes: The study primarily assesses the immediate response to treatment. Including long-term follow-up data could provide insights into the sustainability of the response and overall survival rates. 
  4. Mechanistic Insights & Discussion: While the study speculates on the role of TNFSF9 in the tumor immune microenvironment (TIME), deeper mechanistic studies would strengthen the understanding of how TNFSF9 influences immune responses and therapy outcomes. While your study focuses on predictive biomarkers for immunotherapy efficacy, the article from PMID: 37109725 explores the role of CN in the context of new therapeutic paradigms such as targeted therapy and immunotherapy. This provides a comprehensive view of treatment approaches, enhancing the discussion on personalized treatment strategies in mRCC. Thus, you can cite this article to broaden the scope of therapeutic considerations, linking surgical and pharmacological interventions in the management of mRCC. 
  • To enrich the discussion in your article on TNFSF9 as a biomarker for predicting the efficacy of combination immunotherapy in metastatic renal cell carcinoma (mRCC), you could consider incorporating insights from the article with PMID: 38540735. This article explores the role of MUC1 in RCC, providing a broader context of biomarker dynamics in renal cancer. Please cite and discuss
  1. Figure and Table Clarity: Some figures and tables are dense and could be simplified or presented with clearer explanations to enhance their interpretative value.

Author Response

Reviewer 2:
“Sample Size: The study involves a relatively small cohort of 17 patients. Increasing the sample size could enhance the robustness of the findings and ensure they are more generalizable. If this si not possible please better describe how this limited number of patients could affect your findings.”

Thank you very much for your suggestion. We also have concerns about the small sample size. However, it is difficult to add more samples. Therefore, we have added a description of the possible effects.

Lines 283-287

There are several limitations in the present study. First, we could not exclude the possibility of selection bias due to the retrospective design of the study. Moreover, the sample size is relatively small. As a consequence, only one female was included in the present study. Finally, patient backgrounds were heterogeneous, possibly influencing the results. Further studies would be necessary to rectify these limitations in the future.

“Control Comparisons: While the article focuses on TNFSF9, it would be beneficial to compare its predictive power directly against other potential biomarkers for a more comprehensive evaluation.”

Thank you for your suggestion and we agree with your comment. The biomarker analysis of the CheckMate-214 trial has shown that PD-L1 expression level is a biomarker of the combination therapy with nivolumab and ipilimumab (Motzer RJ et al. J Immunother Cancer. 2022). Therefore, we have evaluated the AUC of PD-L1 in our cohort. The result was added in Figure 3C and we additional describe in the manuscript as below;

Line121

The AUC is also largest for TNFSF9 (AUC=0.9444), whereas that of PD-L1 is lower (AUC=0.75) in the present cohort (Figure 3A, 3C).

“Long-term Outcomes: The study primarily assesses the immediate response to treatment. Including long-term follow-up data could provide insights into the sustainability of the response and overall survival rates.”

Thank you for your valuable remarks. Unfortunately, we did not collect long-term prognosis for each patient, because the purpose of this study was to find a predictive biomarker of response in a combination therapy with nivolumab and ipilimumab. We will refer to this information when we conduct similar studies in the future.

“Mechanistic Insights & Discussion: While the study speculates on the role of TNFSF9 in the tumor immune microenvironment (TIME), deeper mechanistic studies would strengthen the understanding of how TNFSF9 influences immune responses and therapy outcomes. While your study focuses on predictive biomarkers for immunotherapy efficacy, the article from PMID: 37109725 explores the role of CN in the context of new therapeutic paradigms such as targeted therapy and immunotherapy. This provides a comprehensive view of treatment approaches, enhancing the discussion on personalized treatment strategies in mRCC. Thus, you can cite this article to broaden the scope of therapeutic considerations, linking surgical and pharmacological interventions in the management of mRCC.”

Thank you very much for your suggestion. We also consider that a comprehensive view of treatment approach including cytoreductive nephrectomy is important for patients with metastatic renal cell carcinoma. Therefore, I have referred to the article you mentioned and added the description.

Lines 205-207

Cytoreductive nephrectomy (CN) is also a keystone treatment strategy for mRCC. Indeed, several clinical trials are ongoing to elucidate the role of CN in the immunotherapy era [15]. As such, predictive biomarkers might bring significant insight to the appropriate patient selection for CN.

“To enrich the discussion in your article on TNFSF9 as a biomarker for predicting the efficacy of combination immunotherapy in metastatic renal cell carcinoma (mRCC), you could consider incorporating insights from the article with PMID: 38540735. This article explores the role of MUC1 in RCC, providing a broader context of biomarker dynamics in renal cancer. Please cite and discuss”

Thank you very much for your suggestion. Your comment is very valuable. Therefore, we have added a description of MUC1 as a biomarker for renal cancer in the discussion.

Lines 209-212

Biomarkers have been previously reported to predict the efficacy of systemic therapy for mRCC as in the example of transmembrane protein mucin-1 (MUC1), which was high-lighted as a candidate biomarker for antiangiogenic treatment [16]. However, reliable predictive biomarkers for NIVO + IPI remain unidentified for clinical practice.

Figure and Table Clarity: Some figures and tables are dense and could be simplified or presented with clearer explanations to enhance their interpretative value.”

Thank you for making this important point. We have revised figures and Tables to make it easier to read.

Round 2

Reviewer 2 Report

Comments and Suggestions for Authors

the revised manuscript is now worthy of publication